# Transport of antibody into the skin is only partially dependent upon the neonatal Fc-receptor

**Gibran Nasir**[¤], **Photini Sinnis** [iD]*

Johns Hopkins Malaria Institute and Johns Hopkins Bloomberg School of Public Health, Baltimore, Maryland, United States of America

¤ Current address: Harvard Business School, Boston, Massachusetts, United States of America
* psinnis1@jhu.edu

**Data Availability Statement:** All relevant data are within the manuscript and its Supporting information files.

**Funding:** This study was supported by the National Institutes of Health (Grant No. R01 AI132359) awarded to PS and by the Bloomberg Family

## Abstract

The dermis is the portal of entry for most vector-transmitted pathogens, making the host's immune response at this site critical in mitigating the magnitude of infection. For malaria, antibody-mediated neutralization of *Plasmodium* parasites in the dermis was recently demonstrated. However, surprisingly little is known about the mechanisms that govern antibody transport into the skin. Since the neonatal Fc receptor (FcRn) has been shown to transcytose IgG into various tissues, we sought to understand its contribution to IgG transport into the skin and antibody-mediated inhibition of *Plasmodium* parasites following mosquito bite inoculation. Using confocal imaging, we show that the transport of an anti-Langerin mAb into the skin occurs but is only partially reduced in mice lacking FcRn. To understand the relevance of FcRn in the context of malaria infection, we use the rodent parasite *Plasmodium berghei* and show that passively-administered anti-malarial antibody in FcRn deficient mice, does not reduce parasite burden to the same extent as previously observed in wild-type mice. Overall, our data suggest that FcRn plays a role in the transport of IgG into the skin but is not the major driver of IgG transport into this tissue. These findings have implications for the rational design of antibody-based therapeutics for malaria as well as other vector-transmitted pathogens.

## Introduction

Malaria-causing *Plasmodium* parasites are deposited into host skin as infected *Anopheles* mosquitoes probe for blood [1–5]. In order to establish infection, inoculated sporozoites must exit the dermal inoculation site and go to the liver, where they enter hepatocytes and develop into the next life cycle stage. To achieve this goal, sporozoites actively move in the skin to find and enter the blood circulation, which carries them to the liver. Sporozoites can spend up to several hours in the dermis [6, 7], with only a fraction eventually exiting the skin in what can be described as a slow trickle [5–7]. We and others have shown previously that sporozoites are vulnerable to IgG-mediated inhibition in the dermis and observed that inhibitory antibodies

Foundation. GN also received support in the form of the Emergent BioSolutions Fellowship. The Microscopy Facility was supported the by Office of the Director of the National Institutes of Health (Grant No. S10OD016374).

**Competing interests:** The authors have declared that no competing interests exist.

impact sporozoite motility [8, 9]. Importantly, the only malaria vaccine candidate to show efficacy in Phase III clinical trials is RTS,S a subunit vaccine composed of the sporozoite's major surface protein [10], with follow-up studies demonstrating that protection correlates with antibody levels [11, 12]. However, there is a lack of understanding as to how antibody gets into the skin; is it transported and if so how, or does it primarily access this compartment via the rupturing of blood vessels that occurs when vectors seek blood? Since the skin is where the malaria parasite is extracellular for the longest period of time in its mammalian host, it is likely that the dermis is a site of great vulnerability for the malaria parasite. Given the importance of antibodies at the dermal inoculation site in malaria, we sought to better understand the mechanism by which these antibodies are transported from the circulation into the dermis.

Transport of antibody from the blood circulation into tissue can occur by paracellular or transcellular transport [13]. Since antibody molecules are too large to diffuse through most intact endothelial cell barriers, paracellular transport into the skin is only observed in inflamed tissue, when the endothelial cell barrier is disrupted [14]. Transcellular transport can occur via a non-receptor mediated process such as macropinocytosis or by a receptor-mediated process. In the latter case, the most notable receptor is the neonatal Fc receptor (FcRn), a non-canonical FcγR, discovered for its critical role in the transfer of maternal antibodies to the fetus [15, 16]. FcRn is involved in the bidirectional transcytosis of IgG between the blood circulation and the intestinal and upper respiratory mucosa [17] and in the kidney, where IgG is transported across the filtration membrane into the urinary filtrate [18, 19]. FcRn expression has been observed in liver endothelia, brain microvascular endothelia, and the extensive vasculature of the skin [20–22], suggesting that it could be involved in the transport of anti-*Plasmodium* IgG into the dermis. Importantly, FcRn-mediated endocytosis enables IgG to avoid intracellular degradation pathways and leads to its recycling, significantly extending the half-life of serum IgG [16, 23].

Given the paucity of data as to how antibody gets into the dermis, together with the known role of FcRn in IgG transport into tissue and our demonstration that antibody in the skin can significantly decrease *Plasmodium* sporozoite infectivity, we sought to determine the extent to which FcRn is responsible for IgG transport into the dermis. Here we provide evidence that antibody transport into the skin occurs, however, FcRn is not the major pathway by which IgG antibodies enter the dermis.

## Results

To investigate antibody transport into the skin and the role of FcRn in this process, we used mice lacking FcRn and control mice of identical background, both bred in our animal facility. The steady-state presence of a large amount of IgG in the skin makes it technically challenging to investigate the transport of newly introduced IgG into the dermis. Thus, we decided to take an imaging approach, quantifying the staining of Langerhans cells after intravenous inoculation of an anti-Langerin (CD207) monoclonal antibody (mAb). First we verified that this mAb recognized Langerhans cells and that these cells were present in equal numbers in FcRn -/- and wild-type control mice (S1 Fig). Once validated, we intravenously inoculated antibody into wild-type and FcRn -/- mice and excised the ears of the mice 24 hrs later. The ears were fixed, stained with a fluorescently-tagged secondary antibody, and imaged by confocal microscopy (experimental outline shown in Fig 1A). As shown, when increasing amounts of antibody (12.5 µg to 50 µg) were inoculated into wild-type and FcRn -/- mice, the mean sum fluorescence intensity of the stained cells increased in a dose-dependent manner in both wild-type and FcRn -/- mice (Fig 1B). However, at each antibody dose tested, we observed approximately a 50% reduction in the mean sum fluorescence in FcRn -/- mice compared to wild-

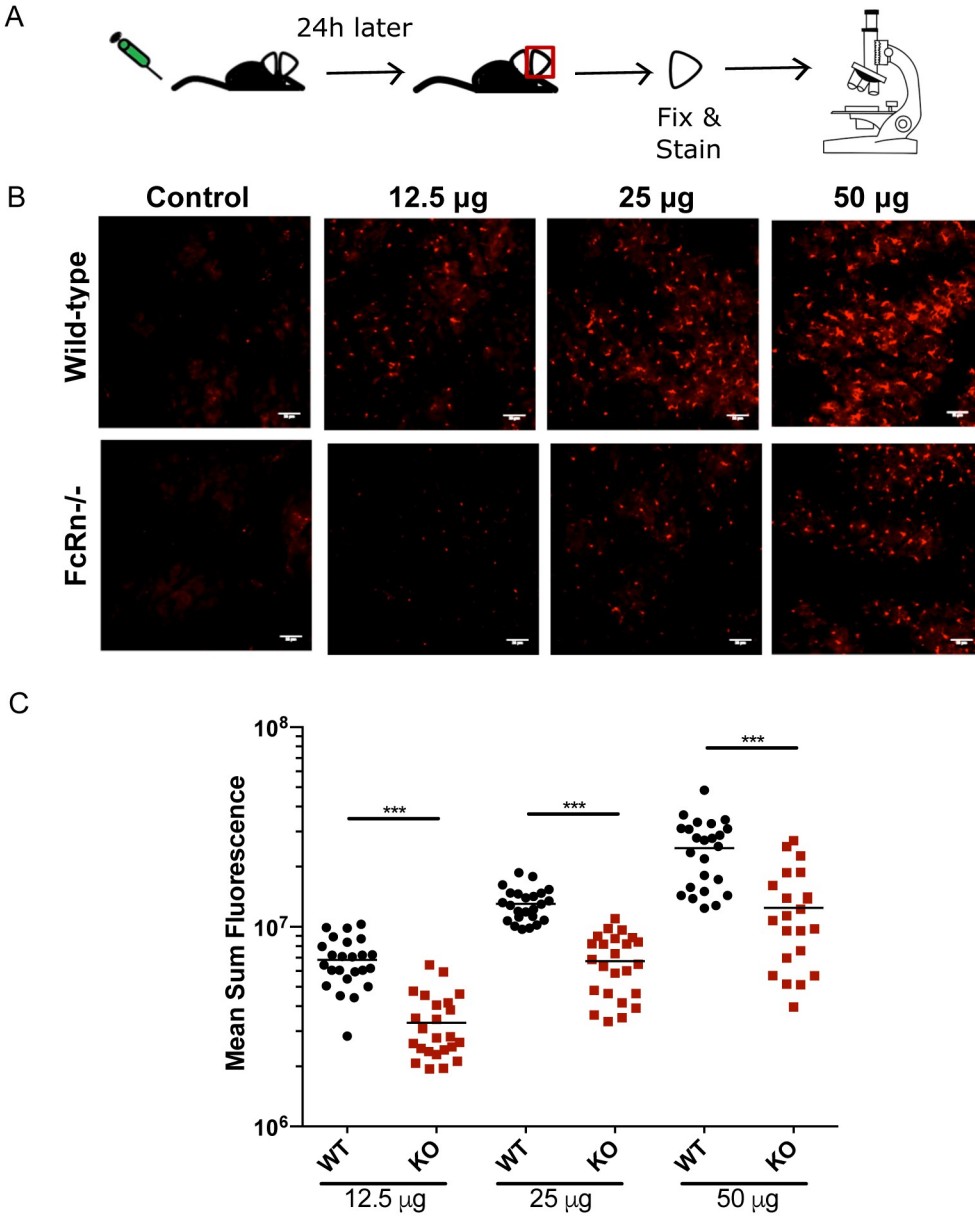

**Fig 1. Concentration-dependent labeling of Langerhans cells in FcRn -/- and wild-type mice. (A)**. Schematic of the experimental workflow: Wild-type and FcRn -/- mice were inoculated IV with the indicated concentrations of an anti-Langerin mAb. Twenty-four hours later, mouse ears were excised, fixed, and stained with a secondary antibody. Controls were not inoculated with anti-Langerin mAb prior to ear removal but were stained with secondary antibody. Ears were imaged by confocal microscopy. **(B)** Representative maximum projection images of Langerhans cell staining in ears of mice IV inoculated with anti-Langerin mAb and control ears (scale bar = 35 μm). **(C)** Dot plot showing the mean sum fluorescence of labeled Langerhans cells in each image, with the line representing the mean. Data from two independent experiments were pooled. Statistical comparisons were performed using two-tailed Mann-Whitney tests (p<0.001). The sum fluorescence was calculated using relative fluorescence units (RFU), with each spot representing the mean sum fluorescence in one image. Identical exposure times were used for all images.

type controls (Fig 1C; p<0.001), suggesting that FcRn does contribute to antibody transport in this location but is clearly not the only mechanism responsible for this process. Importantly, there were no fluorescent cells in the control ears (no antibody) from either wild-type or FcRn -/- mice.

We then evaluated the transport of anti-Langerin antibody at two different time points, inoculating 12.5 μg of anti-Langerin antibody intravenously into wild-type or FcRn deficient mice and harvesting the ears for analysis 12 or 24 hrs after antibody inoculation (experimental outline shown in Fig 2A). Representative images from wild-type and FcRn deficient mice for each time point are shown in Fig 2B. Though Langerhans cells were labeled in both FcRn -/-

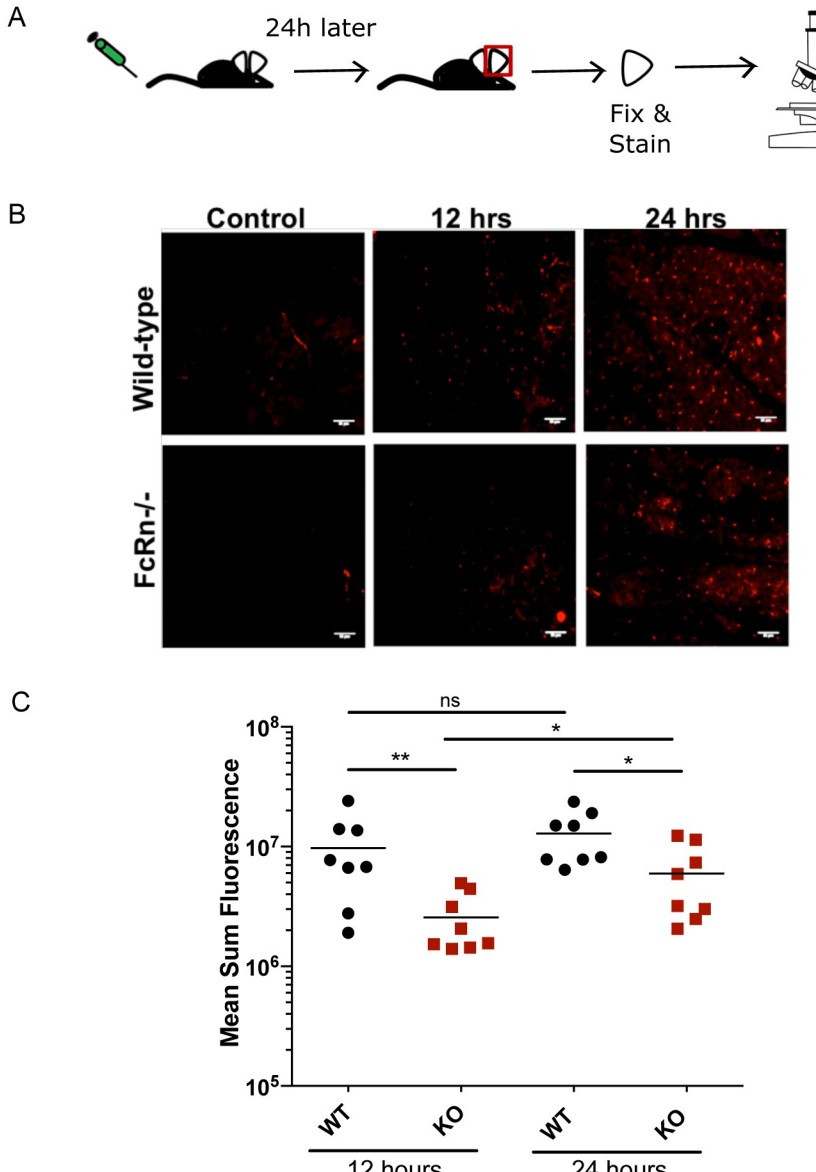

**Fig 2. Kinetics of labeling of Langerhans cells in FcRn -/- mice and wild-type mice. (A).** Schematic of the experimental workflow: Wild-type and FcRn -/- mice were IV inoculated with 12.5 μg of anti-Langerin mAb either 12 hrs or 24 hrs prior to ear harvest. After excision, ears were fixed and stained with a secondary antibody. Controls were not inoculated with anti-Langerin mAb prior to ear removal but were stained with secondary antibody. All ears were imaged using confocal microscopy. **(B)** Representative maximum projection images of Langerhans cell staining in Wild-type and FcRn -/- mice (scale bar = 35 μm). **(C)** Dot plot showing the mean sum fluorescence of labeled Langerhans cells in each image (n = 2 images/ear), with the line representing the mean. Data from two independent experiments were pooled. Statistical comparisons were performed using two-tailed Mann-Whitney tests on the pooled data (* = p<0.05; ** = p<0.01). The sum fluorescence was calculated using relative fluorescence units (RFU), with each spot representing the mean sum fluorescence in one image. Identical exposure times were used for all images.

and wild-type mice at both time points, the fluorescence signal on Langerhans cells in FcRn -/- mice was significantly lower in FcRn -/- mice (Fig 2C). At 12 hrs post-inoculation, we observed a greater difference in Langerhans cell fluorescence between FcRn -/- and wild-type mice, compared to 24 hrs. These data suggest that FcRn may play a role in the more rapid distribution of IgG into the skin. Indeed, the difference between anti-Langerin staining in wild-type mice at 12 and 24 hrs is not significant while in FcRn deficient mice this difference is significant (p<0.05). Overall, the studies outlined above demonstrate that a mAb specific for Langerin can exit the blood circulation and enter the skin and that this process occurs, though less efficiently, in the absence of FcRn.

Next, we wanted to extend these findings to a known inhibitory antibody of rodent malaria sporozoites, mAb 3D11 [24], that we found exerts the majority of its effect at the dermal inoculation site [8]. We first compared the clearance kinetics of serum mAb 3D11 in wild-type and FcRn deficient mice. Previous studies have demonstrated that FcRn is also a recycling receptor and is required for the long serum half-life of IgG [16, 25]. As shown in Fig 3A, when serum antibody concentration is normalized to the respective value in the first sample (at 6 hrs), the clearance kinetics over the first 24 hrs after IV inoculation are not significantly different in wild-type and FcRn deficient mice. However, when non-normalized data are plotted the initial antibody levels in FcRn-/- mice are lower compared to WT controls (Fig 3B). This could be because the FcRn-/- mice were slightly older and possibly larger, or it could be due to increased antibody degradation due to decreased antibody recycling in the FcRn-/- mice. Indeed though the majority of studies on the recycling function of FcRn focus on the degradation of antibody over days, a few studies that included clearance kinetics over the first 24 hrs demonstrate that in FcRn-/- mice clearance of IgG from the circulation is increased in this time frame [26, 27]. In the first 24 hrs post inoculation, serum antibody levels in WT mice decrease rapidly likely because of equilibration with the extravascular compartment, possibly via macropinocytosis, which is likely also occurring in the FcRn-/- mice. In a previous study comparing sporozoite infectivity in passively immunized WT mice challenged by either mosquito bite or IV inoculation, we allowed for antibody equilibration prior to challenge. Thus to compare our current results to that study, we selected 22 hrs after antibody inoculation as the timepoint to challenge the FcRn-/- mice.

Antibody targeting sporozoites can act both in the skin and blood circulation and we have developed a model that allows us to distinguish between these two scenarios, comparing infection in passively-immunized mice after IV versus mosquito bite challenge, where parasite dose by each challenge route is equivalent [8]. Here we used this model to investigate the difference in efficacy of mAb 3D11 in the context of FcRn. Since the wild-type mice used in our previous study [8] were on a 6N background and FcRn deficient mice and the matched controls in this study were on a 6J background, we first compared sporozoite infectivity after IV and mosquito bite inoculation in these two strains. As shown in S2A Fig there was no significant difference in sporozoite infectivity in wild-type 6N and 6J C57Bl/6 mice. As an additional control, we tested whether FcRn deficient mice were equally susceptible to sporozoite infection, after IV and mosquito bite inoculation, and found no difference in sporozoite infectivity between the FcRn deficient mice and wild-type controls (S2B Fig).

We then went on to compare the efficacy of mAb 3D11 on sporozoites inoculated IV or by mosquito bite in FcRn deficient mice. FcRn deficient mice were passively immunized with 25 μg mAb 3D11 or control IgG (mIgG) administered intravenously 22 hrs prior to sporozoite challenge. The mice were then challenged by intravenous inoculation of 350 sporozoites or the bites of 8 infected mosquitoes and forty hours later liver parasite burden was determined (experimental outline shown in Fig 4A). As expected, there was no significant difference in the liver parasite burden between the control groups challenged with sporozoites administered

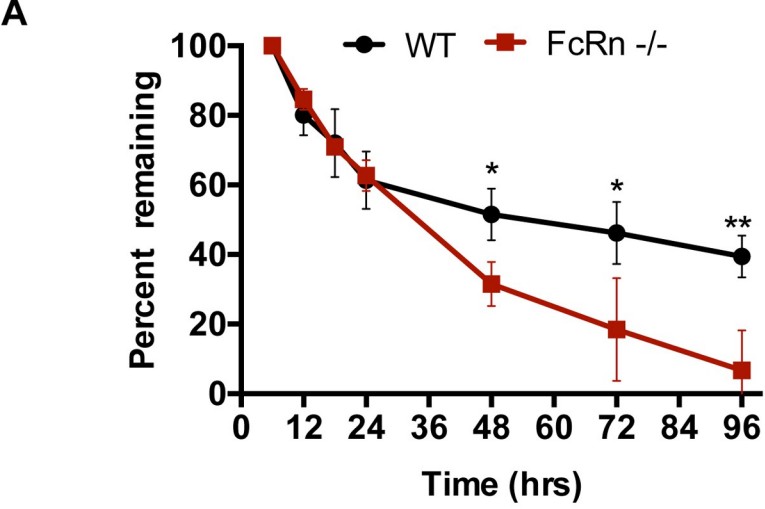

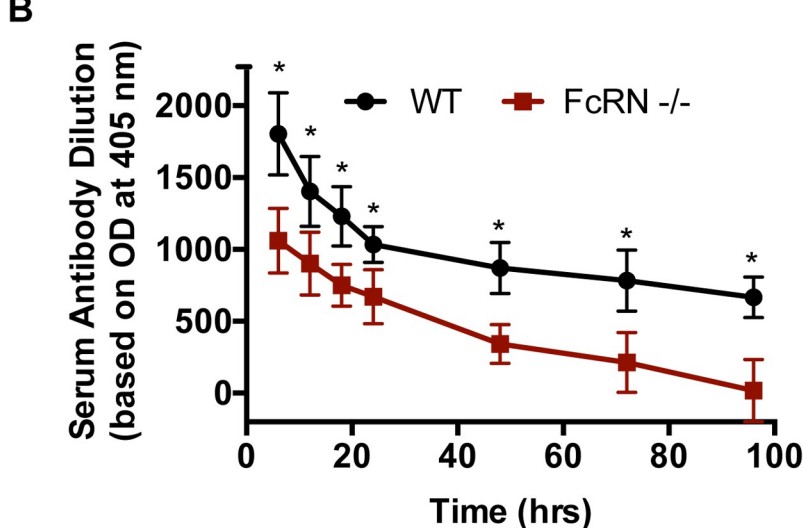

**Fig 3. Average serum clearance kinetics of mAb 3D11 in FcRn -/- and control C57Bl/6J mice.** At the indicated time points following IV inoculation of 50 μg of mAb 3D11, blood was collected retro-orbitally and the level of mAb 3D11 in the serum was determined by ELISA. **(A)** The percentage of mAb 3D11 remaining at each time point in WT and FcRn -/- mice was determined as a ratio with the amount of serum mAb 3D11 at the respective 6 hr value. Shown is the mean ± standard deviation of 3 to 4 mice per time point. **(B)** Non-normalized serum clearance data for WT and FcRn-/- mice is shown. Absorbance was measured in serial dilutions of serum and the dilution corresponding to an absorbance of 1.0 is shown. No comparisons between WT and FcRn-/- mice reached statistical significance. Statistics for both panels compared the percent mAb 3D11 remaining (panel A) or the amount of mAb remaining (panel B), between control and FcRn -/- mice at each time point. Statistical comparisons were performed using an unpaired t-test (two-tailed, α = 0.05).

either intravenously or by mosquito bite (Fig 4B). However, in mice passively-immunized with mAb 3D11, there was greater inhibition of sporozoite infectivity in the mice challenged by mosquito bite (p<0.001), suggesting that even in the absence of FcRn, mAb 3D11 can enter the dermis and is able to inhibit sporozoites in this location. Though similar to our previous findings in wild-type mice, the effect of antibody on mosquito-inoculated sporozoites was not as great in the FcRn deficient mice compared to what we have previously observed in wild-type mice [8]. This comparison is shown in Fig 4C: The average fold-reduction between

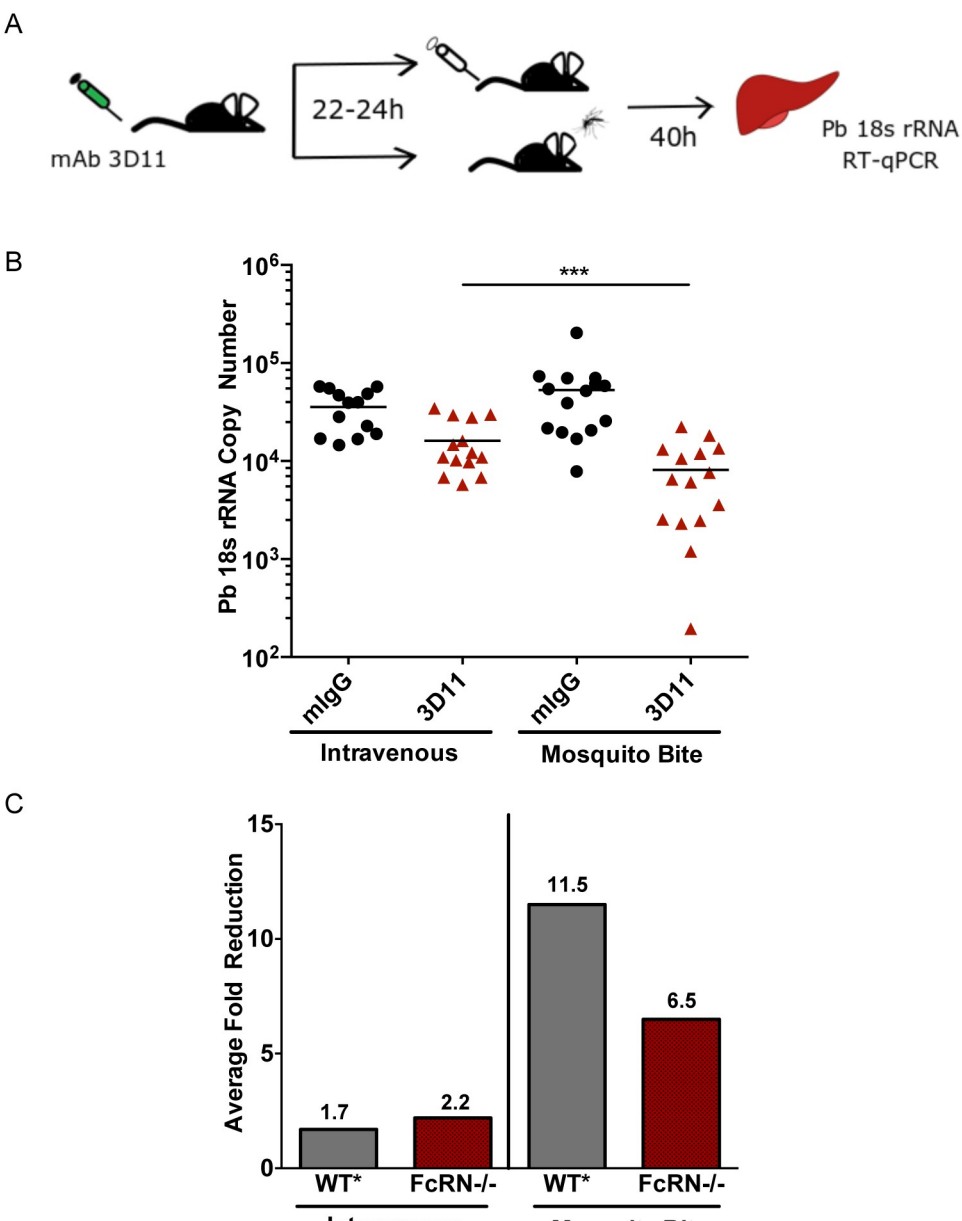

**Fig 4. Efficacy of mAb 3D11 against IV and mosquito bite inoculated sporozoites in FcRn -/- mice. (A)**. Schematic of the experimental workflow: FcRn -/- mice were inoculated IV with 25 µg of mAb 3D11, specific for the major surface protein of sporozoites, or with an equivalent amount of mouse IgG. The next day, mice were challenged with 350 sporozoites inoculated IV or by 8 infected mosquito bites. 40 hrs post-sporozoite inoculation liver parasite burden was determined. **(B)**. Dot plot showing liver parasite burdens of mAb 3D11 and control IgG inoculated FcRn -/- mice, challenged by either IV or mosquito bite inoculation of sporozoites. Data are pooled from 3 independent experiments (n = 4–5 mice/group). No statistically significant difference was found between the control groups (mIgG groups IV vs MB). The difference between the mAb 3D11 groups IV vs MB was statistically significant (p<0.001). Statistical comparisons were performed using a multi-level mixed effects model. **(C)** The average fold-reduction between control and mAb 3D11 groups, in both the IV challenged and mosquito bite challenged mice was calculated by taking the negative inverse of the ratio [-1/(3D11:mIgG)] of the average *P.berghei* 18s copy number. The fold-reduction in FcRn-/- mice, IV and mosquito bite challenged, is calculated using the data in panel B. The fold-reduction for the same concentration of mAb 3D11 in wild-type (WT*) mice is from a previous publication [9] and included for comparison.

control and mAb 3D11 groups in FcRn deficient mice challenged by mosquito bite was 6.5 fold whereas in wild-type mice it was 11.5 fold lower [8]. In contrast, after IV challenge of sporozoites, the average fold reduction between control and mAb 3D11 groups was similar, 2.2 versus 1.7 in the FcRn deficient and wild-type mice, respectively.

## Discussion

Given the enormous global health significance of mosquito-borne infections and bacterial infections of the skin, there is surprisingly little clarity around mechanisms governing homeostatic IgG deposition in the skin. To understand the role of FcRn in antibody deposition into dermal tissue, we selected an antibody targeting Langerhans cells as they provide a relatively fixed antibody target in the dermis since they self-renew in place [28]. We then imaged binding of IV-inoculated antibody to these cells in FcRn deficient and wild-type mice. The skin microvasculature is composed of endothelial cells that are connected to one another by either adherens or tight junctions, which during inflammation or injury can become permeable to paracellular diffusion of immunoglobulins into the dermis [29]. However, at steady state, IgG deposition into the dermis through passive diffusion seems unlikely due to its large size. A more likely alternative is transcellular transport, either receptor mediated or non-receptor mediated (such as fluid phase) transcytosis–or both. Our data show that FcRn has a minor role in this process, indicating that fluid phase transcytosis of antibody is likely the dominant mechanism by which IgG moves into the skin.

Though we found a small role for FcRn, a recent study by Ono et al. found no role for FcRn in the deposition of AK18, an IgG1 autoantibody, in the epidermis of mice [30]. There are, however, some differences in the way the two studies were performed which may account for these differences. Ono et al. used β2m-/- mice and only looked at antibody in the skin at 6 hrs post-inoculation, while we used FcRn-/- mice and performed our assessments at 12 and 24 hrs after antibody inoculation. The different mutants in each study could give rise to different results, with β2m-/- mice bearing additional defects compared to FcRn deficient mice. Indeed, it has been reported that IgG half-life in β2m-/- mice is shorter than in FcRn-/- mice, though the reason for this discrepancy is not clear [31]. Additionally, the different time points chosen in the two studies likely impact the results. We chose 12 and 24 hrs post-inoculation with the assumption that IV inoculated antibody equilibrates with the extravascular space in this time frame. In contrast, the Ono study used 6 hrs post-inoculation because they found significant decreases in serum levels of inoculated antibody at 24 hrs.

Results from our study clearly indicate that antibody is transported into the skin. However, this is via one or more mechanism(s) that are independent of FcRn. This can include clathrin-mediated endocytosis, caveolin-dependent endocytosis, or macropinocytosis. In an *in vitro* study performed with human intestinal epithelial cells, partial inhibition in FITC-IgG uptake was only observed when an inhibitor targeting macropinocytosis was used, with no significant difference in FITC-IgG uptake when inhibitors of either clathrin- or caveolin-dependent endocytosis were used [32]. The Ono study above, which was specifically looking at antibody transport into the skin, found that caveolin-dependent endocytosis was an important mechanism by which antibody is transported into the skin [30]. Therefore, it seems that to some degree, both macropinocytosis as well as caveolin-dependent endocytosis are involved in IgG internalization and transcytosis, though the extent and tissue-specificity of these processes is not known. While future research will help clarify the precise mechanisms involved in IgG transcytosis, it is not surprising that redundant mechanisms should be at play.

Overall, our data demonstrate that FcRn may have some role in antibody transport into the skin, though other transport mechanism(s) also function in this process. This is suggested by

two observations: 1) the decreased amount of anti-langerin IgG in the skin of FcRn-/- mice, as is evidenced by the decreased staining of Langerhans cells, and 2) the decreased amount of mAb 3D11 in the skin, which is suggested by the decreased inhibitory activity on sporozoites observed in FcRn-/- mice. These findings could be due to increased destruction of antibody due to the lack of FcRn or to the decreased transfer of antibody into the skin via FcRn, or both. In either case, given our current findings and previous studies demonstrating an important role of dermal antibody in protection against malaria sporozoites [8, 9], it is possible that increasing the half-life of these antibodies could improve current vaccine efforts targeting sporozoites. Through a rational design of IgG-based therapeutics, the half-life of molecules such as antibodies and Fc-drug conjugates can be prolonged in the serum by several fold, and the amount deposited into tissues significantly enhanced [33–35] and reviewed in [36]. These ideas should be applicable to drug and inhibitory antibody design for malaria as well. For instance, antibodies and other biologics targeting *Plasmodium* sporozoites could be engineered for enhanced half-life and transcytosis into dermal tissue, possibly via an FcRn-mediated mechanism, thus increasing the efficacy of malaria vaccines targeting *Plasmodium* sporozoites.

## Materials & methods

### Animals

FcRn -/- (Jackson stock #003982; [16]) and C57BL/6J wild-type controls were purchased from Jackson Laboratories (Bar Harbor, ME) and bred in the animal facility at the Johns Hopkins Bloomberg School of Public Health. Female C57Bl/6N mice for S2 Fig were purchased from Taconic Farms (Derwood, MD). For experiments, mice were age matched, and only female mice were used. All animal work was performed in accordance with the Animal Care and Use Committee (ACUC) guidelines (Protocol #M017H325). For ear removal, serum sampling, and liver harvest, mice were aneasthetized with ketamine/xylazine until non-responsive to pain and then sacrificed by cervical dislocation after the procedure.

### Parasites and mosquitoes

*Anopheles stephensi* mosquitoes were reared in the insectary at the Johns Hopkins Bloomberg School of Public Health and infected with wild-type *Plasmodium berghei* ANKA parasites. Infected mosquitoes were used on days 18 to 24 post-infectious bloodmeal.

### Quantification of Langerhans cell labeling

Mice were inoculated IV with 12.5, 25, or 50 μg Rat anti-Langerin mAb (Isotype IgG2a; #14-2075-82, Thermo Fisher). At 12h or 24h post-antibody administration, mouse ears were excised and carefully split in order to obtain dorsal leaflets without cartilage. The dorsal leaflets were fixed overnight at 4°C degrees in freshly prepared PLP buffer (1% PFA, 0.2M L-lysine, and 0.2% $NaIO_4$ in 0.1M phosphate buffer, pH 7.4). The following day, the leaflets were washed 3 times in 1 mL PBS for 1 hour per wash, and permeabilized and blocked overnight at 4°C in 250 μL Blocking Buffer (0.3% Triton X-100, 0.5% BSA, 10% goat serum in PBS). The leaflets were then incubated for 2 days in the dark at 4°C with goat anti-Rat IgG Cross-adsorbed Alexafluor 594 secondary antibody dilution 1:500 (#A11007, Thermo Fisher Scientific), washed 3 times in 1 mL PBS (pH 7.4) and mounted dermal side up using Prolong gold anti-fade mounting reagent (#P10144, Thermo Fisher Scientific). The ears were imaged within 1–2 days post-mounting using a Zeiss 710 Confocal microscope with a 25x water objective. Each Z-stack consisted of 10 slices, with a total stack depth of 25 μm. Langerhans cells were detected using the spot detection feature (spot diameter of 15 μm and spot depth of 13 μm) in

Imaris software v9.12 and the sum fluorescence intensity measured using relative fluorescence units (RFU) for each spot quantified. This is the fluorescence signal detected by the instrument. Identical exposure times were used for all images. Cells were identified using the spots algorithm wizard, which was applied to the entire imported image stack. Under the parameters for the spots algorithm wizard, background subtraction was included, and the 'quality' filter was used to select as many fluorescent cells as possible. The image was then manually examined in order to ensure that false positives such as hair follicles were not included and removed from the analysis accordingly. The sum fluorescence intensity for the detected spots was exported and analyzed in GraphPad Prism 6 software.

## Antibody serum clearance kinetics

50 μg mAb 3D11 (isotype IgG1) was intravenously injected into 7 to 8 week old wild-type or FcRn -/- mice and 50 μL of blood were collected retro-orbitally under anesthesia (35–50 mg/ kg ketamine and 6–10 mg/kg xylazine) at 6h, 12h, 18h, 24h, 48h, 72h, and 96h following antibody administration. Samples were centrifuged at 2000 x g for 10 minutes (4˚C), and the supernatant used for quantifying serum antibody levels by ELISA. Briefly, 96-well plates (ThermoFisher Scientific, #3455) were coated with 1.0 μg/mL of synthetic *P. berghei* CSP repeat peptide (Multiple Antigen Peptide: 4X PPPPNPNDPPPPNPND–KQIRDSITEEWS) overnight at 4˚C. The following day, the wells were washed 3 times with wash buffer (0.05% Tween 20 in PBS) and blocked with blocking buffer (2% BSA, 0.05% Tween 20 in PBS) for 1 hour at 37˚C. Serial dilutions of each serum sample were prepared in blocking buffer and added to the wells for 1 hour at 37˚C. The wells were washed 3 times and incubated with goat anti-mouse HRP-conjugated secondary antibody (KPL, #074–1806) diluted 1:2000 in blocking buffer for 1 hour at 37˚C. Following 3 washes with wash buffer and 2 washes with PBS (pH 7.4), the plates were developed using a commercially available kit (KPL, #50-62-00) for 5 min and the absorbance measured at 405 nm. In order to normalize the ELISA data across all samples, a standard curve was generated for each sample. This was performed by serially diluting each serum sample 1:50, 1:100, 1:200, 1:400, 1:800, and 1:1600 in blocking buffer. Absorbance was measured at each of these dilutions and graphed. The serum dilution corresponding to an absorbance measurement of 1.0 (a point at which the concentration of the antibody in the serum is directly proportional to the absorbance) was interpolated from the standard curve. These interpolated values were plotted against time to observe the change in serum mAb 3D11 over time.

## Sporozoite challenge experiments

These were performed as described previously [9]. Briefly, sporozoites used in IV challenge were obtained by mosquito dissection in cold L-15 media (#11415064, Thermo Fisher Scientific) and sporozoites were quantified using a hemocytometer. A working solution of 350 sporozoites per 200 μL RPMI 1640 (#22400089, Gibco) was prepared and inoculated into mice via the tail vein. For mosquito bite infection, mosquitoes were transferred into feeding tubes the day prior to challenge and fed only water until ~6 hr prior to challenge at which point they were starved until the experiment. Mice were anesthetized with ketamine-xylazine (35–50 mg/kg ketamine and 6–10 mg/kg xylazine) and placed on a warming plate set to 35˚C. Mouse ears were exposed to a total of 8 mosquito bites (4 bites per ear). The number of bites was ensured by visually supervising the biting process. Forty hours post-infection, mice were aneasthetized and their livers were harvested for quantification of liver parasite burden as outlined previously in [37]. Briefly, livers were homogenized in Trizol (#TR118, Molecular Research Center), total RNA was extracted and reverse transcription was performed using

1.5 μg of total RNA and random hexamers (Thermo Fisher Scientific). Following this, *P. berghei* 18S rRNA copy number was quantified by qPCR using primers specific for *P. berghei* 18S rRNA gene [38] and a standard curve generated from 10-fold serial dilutions ($10^7$ to $10^2$) of a plasmid containing the *P. berghei* 18S rRNA amplified fragment. The sequence for the amplifying primers was `Pb-18S-rRNA-5' GGAGATTGGTTTTGACGTTTATGTG` and `Pb-18S-rRNA-3' AAGCATTAAATAAAGCGAATACATCCTTA`.

## Statistical analysis

Statistical comparisons of the mean sum fluorescence intensity in wild-type and FcRn -/- animals for each concentration (Fig 1C) and timepoint (Fig 2C) were done using non-parametric Mann-Whitney tests (two-tailed, $\alpha = 0.05$). Similarly, Mann-Whitney tests were used for determining statistical significance in S2 Fig. An unpaired t-test (two-tailed, $\alpha = 0.05$) was used for determining significance in the percent of remaining antibody between wild-type and FcRn deficient mice at 48h, 72h, and 96h in Fig 3. Statistical comparisons of *P. berghei* 18S rRNA copy number between control and passively immunized mice in Fig 4B were performed using a linear multilevel mixed-effects model, accounting for experimental variation and interaction of treatment and route, using STATA version 13.0 software.

## Supporting information

**S1 Fig. Staining of Langerhans cells in FcRn -/- and control mice.** Wild-type and FcRn -/- mice were inoculated IV with 50 μg of anti-Langerin mAb. Twenty-four hours later, mouse ears were excised, fixed, and stained with a secondary antibody and ears were imaged by confocal microscopy. (A) Maximum projection images of stained Langerhans cells from control and FcRn -/- mice. (B) Total number of stained Langerhans cells in control and FcRn -/- mouse ears counted using the spot function in Imaris. In contrast to the data shown in Figs 1 and 2, which shows fluorescence intensity, these data show total number of spot counts and indicate that there is no difference in the total number of Langerhans cells between Wild-type and FcRn -/- mice. Statistical comparison was performed using an unpaired t-test (two-tailed, $\alpha = 0.05$).
(PDF)

**S2 Fig. Sporozoite infectivity in C57Bl/6J and C57Bl/6N mice and C57Bl/6J wild-type and FcRn -/- mice.** (A) *P. berghei* sporozoite infectivity in C57Bl/6J and C57Bl/6N mice after intravenous or mosquito bite inoculation of parasites. Mice were intravenously inoculated with 350 *P. berghei* sporozoites or exposed to 8 infected mosquito bites and 40 hours later, livers were harvested and the level of Pb 18s rRNA in the liver quantified by RT-qPCR. Shown are the pooled data from two independent experiments (n = 4–5 mice per experiment). Statistical comparisons between C57Bl/6J and C57Bl/6N mice were performed on the pooled data using two-tailed Mann-Whitney tests. The difference between the groups was not statistically significant (ns, p>0.05). (B) *P. berghei* sporozoite infectivity in C57Bl/6J wild-type and FcRn -/- mice after intravenous and mosquito bite inoculation of parasites. Mice were intravenously inoculated with 350 *P.berghei* sporozoites or exposed to 7 infected mosquito bites. Forty hours later, livers were harvested and the level of Pb 18s rRNA in the liver was quantified using RT-qPCR. Shown is data from one experiment (n = 3–4 mice per group). Statistical comparisons between wild-type and FcRn -/- mice were performed using two-tailed Mann-Whitney tests and were not significant (ns, p>0.05).
(PDF)

## Acknowledgments

We would like to thank Dr. Andrea Radtke for her helpful advice in setting up the immunofluorescence assays and Dr. Sachie Kanatani and Sharon Patray for their intellectual contributions throughout the project. We are very grateful to Chris Kizito, Godfree Mlambo, and Abhai Tripathi the team of the parasitology and insectary core facilities at the Johns Hopkins Malaria Research Institute for their outstanding work. We acknowledge the Johns Hopkins School of Medicine Microscopy Facility (MicFac) and thank Dr. Scott Kuo, Barbara Smith, Hoku West-Foyle, and LaToya Roker for their invaluable assistance.

## Author Contributions

**Conceptualization:** Gibran Nasir, Photini Sinnis.

**Data curation:** Gibran Nasir, Photini Sinnis.

**Formal analysis:** Gibran Nasir, Photini Sinnis.

**Funding acquisition:** Photini Sinnis.

**Investigation:** Gibran Nasir.

**Methodology:** Gibran Nasir.

**Supervision:** Photini Sinnis.

**Validation:** Photini Sinnis.

**Visualization:** Gibran Nasir, Photini Sinnis.

**Writing – original draft:** Gibran Nasir, Photini Sinnis.

**Writing – review & editing:** Gibran Nasir, Photini Sinnis.

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
