## [Decision Letter · Decision Letter 0]

28 Sep 2022

PONE-D-22-23114Transport of Antibody into the Skin is Only Partially Dependent Upon the Neonatal Fc-ReceptorPLOS ONE

Dear Photini,

Thank you for submitting your manuscript to PLOS ONE. After careful consideration, we feel that it has merit but does not fully meet PLOS ONE’s publication criteria as it currently stands. Therefore, we invite you to submit a revised version of the manuscript that addresses the points raised during the review process.

The two reviewers raised comments that should be straightforward to address, without the need for additional experimental work. I also noted that one sentence in the abstract is misleading and should be revised ("the protective efficacy of a passively-administered antibody is reduced in the skin in FcRn deficient mice, but not to the same extent that we have previously observed..." (Protection is reduced as compared to WT).

We look forward to receiving your revised manuscript.

Kind regards,

Olivier Silvie, M.D., Ph.D.

Academic Editor

PLOS ONE

Journal Requirements:

"PS: National Institutes of Health R01 AI132359 and Bloomberg Family Philanthropies

GN: Emergent BioSolutions Fellowship

Microscopy Facility: Office of the Director of the National Institutes of Health S10OD016374"

Reviewers' comments:

Reviewer's Responses to Questions

**Comments to the Author**

1. Is the manuscript technically sound, and do the data support the conclusions?

Reviewer #1: Partly

Reviewer #2: Yes

2. Has the statistical analysis been performed appropriately and rigorously? 

Reviewer #1: Yes

Reviewer #2: Yes

3. Have the authors made all data underlying the findings in their manuscript fully available?

Reviewer #1: Yes

Reviewer #2: Yes

4. Is the manuscript presented in an intelligible fashion and written in standard English?

Reviewer #1: Yes

Reviewer #2: Yes

5. Review Comments to the Author

Reviewer #1: The manuscript entitled « Transport of Antibody into the skin is only partially dependent upon the Neonatal Fc-Receptor (FcRn) » by Nasir and Sinnis describes the role of FcRn in the transport of rat IgG2a and mouse IgG1from the blood into the skin by respectively measuring antibody binding to skin cells and neutralization of sporozoites in the skin of WT and FcRn -/- mice.

The authors consistently observed less staining of langerin+ cells in the skin of FcRn-/- mice 12-24h after IV injection of rat anti-langerin IgG2a mAb suggesting that FcRn partly participates in the mAb transport from the blood to the skin.

Accordingly, 22-24h after IV transfer of mouse anti-PbCSP IgG1mAb (3D11), a ~ 7-fold reduction in liver infection of FcRn-/- mice in comparison to ~12-fold decrease in WT mice from (ref 8) was observed after mosquito bite challenge. Equivalent IV challenge produced similar level of reduction in liver infection of passively immunized mice (2.2-fold reduction in FcRn-/- mice vs 1.7 in WT mice (from ref 8)).

Altogether, skin cell binding and sporozoite neutralization in the skin data suggest that FcRn partially contributes to the transport of antibodies from the blood into the skin.

However, since FcRn -/- mice also present faster clearance of plasma antibodies, this phenotype could explain the observed differences. To address this, the authors performed the quantification of the normalized % of antibody remaining in circulation in WT and FcRN -/- mice following passive transfer of mAbs (Fig. 3).

There are two points regarding figure 3 that still need explanation before the acceptation of the manuscript.

The first is that in figure 3, values are determined as a percentage of the amount of 3D11 mAb in the serum at 6h. It is not clear if all points are normalized to the value of WT group at 6h or by the respective 6h values. In the last case,

despite the same normalized kinetics until 24h, it is not possible to appreciate the decrease in serum concentration that could help to explain the differences observed in figures 1, 2 and 4. Please plot the non-normalized data.

The second is that a quick search about mAb clearance curve in FcRn-/- mice showed that at a substantial decrease in the amount of antibodies can be observed at 24h (PMID: 27610650, PMID: 28613103). Please compare/discuss your data with published data.

Reviewer #2: This is a short and simple study assessing a potential role for FcRn-mediated transport into the skin for antibody-mediated protection against Plasmodium sporozoite infection. Using FcRn deficient mice, the authors quantify antibody trafficking into the skin by imaging anti-Langerhans antibodies, finding that trafficking is reduced in deficient mice, but not eliminated. Further, extended time indicates an antibody accumulation as time goes on. The mice were then used in challenge experiments, finding that in mosquito bite challenge, where skin-resident antibodies are critical to protection, protection was reduced in the absence of FcRn. IV challenge saw little or no protection in either background, as expected. Overall, this is a relatively simple experiment that shows a role for FcRN, but also indicates that it is not critical.

Given the simplicity of the study, there is not a lot of major feedback, and the results are straightforward. However, it would be good to address why the authors did not use FcRn binding site knockouts to test this from the antibody side. That would likely be a cleaner way to address this, by knocking out FcRN binding of the antibody, and may have fewer pleiotropic effects than the genetic mouse knockout. Perhaps a discussion of this would suffice, but it is an obvious question an erudite reader will ask.

Minor points:

All log scale graphs lack minor ticks

bottom of 2C was cutoff in the review copy

Fluorescence units were not listed or labeled, or defined in the M/M

6. PLOS authors have the option to publish the peer review history of their article (what does this mean?). If published, this will include your full peer review and any attached files.

Reviewer #1: No

Reviewer #2: No

---

## [Author Response · Author response to Decision Letter 0]

7 Mar 2023

We thank the Reviewers and Editor for their helpful comments on our manuscript. They raised good points, which we have addressed as outlined below.

Editor Comment:

I also noted that one sentence in the abstract is misleading and should be revised ("the protective efficacy of a passively-administered antibody is reduced in the skin in FcRn deficient mice, but not to the same extent that we have previously observed..." (Protection is reduced as compared to WT).

We’re not exactly sure what needed to be changed but think it's the reference to antibody in the skin. We have changed this sentence as follows below but if we misinterpreted please let us know and we will change accordingly.

To understand the relevance of FcRn in the context of malaria infection, we use the rodent parasite Plasmodium berghei and show that the protective efficacy of a passively-administered anti-malarial antibody is reduced in FcRn deficient mice, but not to the same extent as previously observed in wildtype mice.

Reviewer #1: 

There are two points regarding figure 3 that still need explanation before the acceptation of the manuscript.

The first is that in figure 3, values are determined as a percentage of the amount of 3D11 mAb in the serum at 6h. It is not clear if all points are normalized to the value of WT group at 6h or by the respective 6h values. In the last case, despite the same normalized kinetics until 24h, it is not possible to appreciate the decrease in serum concentration that could help to explain the differences observed in figures 1, 2 and 4. Please plot the non-normalized data.

The values are normalized to the respective 6 hr values and this is now made clear in both the Figure 3 legend and the Results section:

Fig 3 legend: The percentage of mAb 3D11 remaining at each time point in WT and FcRn -/- mice was determined as a ratio with the amount of serum mAb 3D11 at the respective 6 hr value.

Results: As shown in Figure 3A, when serum antibody concentration is normalized to the respective value in the first sample (at 6 hrs)…

Thus, the reviewer is correct that any differences in clearance kinetics at early time points between in WT vs FcRn KO mice prior to 24 hrs might not be evident. We plotted the data this way because initial sampling can give variable results dependent upon mouse size and other factors, and we found that serum clearance studies are frequently reported this way (References 1-4 below). However, we agree with the reviewer that its good to show the non-normalized data which indicate, as the reviewer predicted, that the decreased serum concentration in the FcRn mice at the early time points could explain the differences observed in Figures 1, 2, and 4. We now include both normalized and non-normalized data in the revised Figure 3. We have also changed the text in several places to reflect this: Results section (page 5), legend for Figure 3, and Discussion (bottom of page 8).

Fig 3 legend: (B) Non-normalized serum clearance data for WT and FcRn-/- mice is shown. Absorbance was measured in serial dilutions of serum and the dilution corresponding to an absorbance of 1.0 is shown. No comparisons between WT and FcRn-/- mice reached statistical significance.

Results: However, when non-normalized data are plotted the initial antibody levels in FcRn-/- mice are lower compared to WT controls (Fig 3B). This could be because the FcRn-/- mice were slightly older and possibly larger, or it could be due to increased antibody degradation due to decreased antibody recycling in the FcRn-/- mice. Indeed though the majority of studies on the recycling function of FcRn focus on the degradation of antibody over days, a few studies that included clearance kinetics over the first 24 hrs demonstrate that in FcRn-/- mice clearance of IgG from the circulation is increased in this time frame [26, 27]. In the first 24 hrs post inoculation, serum antibody levels in WT mice decrease rapidly likely because of equilibration with the extravascular compartment, possibly via macropinocytosis, which is likely also occurring in the FcRn-/- mice. In a previous study comparing sporozoite infectivity in passively immunized WT mice challenged by either mosquito bite or IV inoculation, we allowed for antibody equilibration prior to challenge. Thus to compare our current results to that study, we selected 22 hrs after antibody inoculation as the timepoint to challenge the FcRn-/- mice.

Discussion: Overall, our data demonstrate that FcRn may have some role in antibody transport into the skin, though other transport mechanism(s) also function in this process. This is suggested by two observations: 1) the decreased amount of anti-langerin IgG in the skin of FcRn-/- mice, as is evidenced by the decreased staining of Langerhans cells, and 2) the decreased amount of mAb 3D11 in the skin, which is suggested by the decreased inhibitory activity on sporozoites observed in FcRn-/- mice. These findings could be due to increased destruction of antibody due to the lack of FcRn or to the decreased transfer of antibody into the skin via FcRn, or both. In either case, given our current findings and previous studies demonstrating an important role of dermal antibody in protection against malaria sporozoites [8,9] , it is possible that increasing the half-life of these antibodies could improve current vaccine efforts targeting sporozoites.

The second is that a quick search about mAb clearance curve in FcRn-/- mice showed that at a substantial decrease in the amount of antibodies can be observed at 24h (PMID: 27610650, PMID: 28613103). Please compare/discuss your data with published data.

We actually did not mean to imply that antibody levels do not decrease in the first 24 hrs. In fact in both WT and FcRn-/- mice this is a time of greatest decrease in serum levels largely because of antibody equilibration with the extravascular spaces. However, in line with what the reviewer is saying, when we plot the non-normalized data, its clear that there is less serum antibody in FcRn-/- mice even at early time points. We now have added the following sentence to the Results section (page 5) and cite additional references (References 5 and 6 below and in paper new References 26 and 27) to reflect this:

Indeed though the majority of studies focused on the recycling function of FcRN focus on the degradation of antibody over days, a few studies that included clearance kinetics over the first 24 hrs demonstrate that in FcRN-/- mice clearance of IgG from the circulation is increased in FcRN-/- in this time frame [26, 27].

Reviewer #2: It would be good to address why the authors did not use FcRn binding site knockouts to test this from the antibody side. That would likely be a cleaner way to address this, by knocking out FcRN binding of the antibody, and may have fewer pleiotropic effects than the genetic mouse knockout. Perhaps a discussion of this would suffice, but it is an obvious question an erudite reader will ask.

One can digest antibodies with papain and obtain Fab fragments that do not contain the Fc portion. Though many experiments have been done with these Fab fragments, when they are inoculated into the blood circulation, they are rapidly cleared (Reference 7 below). Indeed in the case of mAb 3D11 that we used in this paper, a previous study testing Fab fragments had to preincubate sporozoites with the Fabs in vitro prior to IV inoculation specifically because they were cleared so quickly once in the blood that they could not be tested by passively immunizing the mice (Reference 8 below). Nonetheless, there are now many studies in which, the Fc regions are changed to investigate binding to the FcRN receptor. We have now included additional references citing these studies in the Discussion (Reference 9 & 10 below which are added as References 33 and 34 in the paper):

Through a rational design of IgG-based therapeutics, the half-life of molecules such as antibodies and Fc-drug conjugates can be prolonged in the serum by several fold, and the amount deposited into tissues significantly enhanced (33, 34, 35 and reviewed in [36]).

Minor points:

All log scale graphs lack minor ticks

Log scale graphs now have minor ticks

bottom of 2C was cutoff in the review copy

We have tried to correct this

Fluorescence units were not listed or labeled, or defined in the M/M 

We have now corrected this in the Methods and Figure legends:

Methods: … and the sum fluorescence intensity measured using relative fluorescence units (RFU) for each spot quantified. This is the fluorescence signal detected by the instrument. Identical exposure times were used for all images.

Fig 1 and 2 Legends: The sum fluorescence was calculated using relative fluorescence units (RFU), with each spot representing the mean sum fluorescence in one image. Identical exposure times were used for all images.

References

1) Montoyo HP, Vaccaro C, Hafner M, Ober RJ, Mueller W, Ward ES. Conditional deletion of the MHC class I-related receptor FcRn reveals the sites of IgG homeostasis in mice. Proc Natl Acad Sci U S A. 2009 Feb 24;106(8):2788-93. doi: 10.1073/pnas.0810796106. PMID: 19188594.

2) Roopenian DC, Christianson GJ, Sproule TJ, Brown AC, Akilesh S, Jung N, Petkova S, Avanessian L, Choi EY, Shaffer DJ, Eden PA, Anderson CL. The MHC class I-like IgG receptor controls perinatal IgG transport, IgG homeostasis, and fate of IgG-Fc-coupled drugs. J Immunol. 2003 Apr 1;170(7):3528-33. doi: 10.4049/jimmunol.170.7.3528. PMID: 12646614.

3) Junghans RP, Anderson CL. The protection receptor for IgG catabolism is the beta2-microglobulin-containing neonatal intestinal transport receptor. Proc Natl Acad Sci U S A. 1996 May 28;93(11):5512-6. doi: 10.1073/pnas.93.11.5512. PMID: 8643606

4) Israel EJ, Wilsker DF, Hayes KC, Schoenfeld D, Simister NE. Increased clearance of IgG in mice that lack beta 2-microglobulin: possible protective role of FcRn. Immunology. 1996 Dec;89(4):573-8. doi: 10.1046/j.1365-2567.1996.d01-775.x. PMID: 9014824

5) Montoyo HP, Vaccaro C, Hafner M, Ober RJ, Mueller W, Ward ES. Conditional deletion of the MHC class I-related receptor FcRn reveals the sites of IgG homeostasis in mice. Proc Natl Acad Sci U S A. 2009 Feb 24;106(8):2788-93. doi: 10.1073/pnas.0810796106. PMID: 19188594.

6) Eigenmann MJ, Fronton L, Grimm HP, Otteneder MB, Krippendorff BF. Quantification of IgG monoclonal antibody clearance in tissues. MAbs. 2017 Aug/Sep;9(6):1007-1015. doi: 10.1080/19420862.2017.1337619. PMID: 28613103

7) SPIEGELBERG HL, WEIGLE WO. THE CATABOLISM OF HOMOLOGOUS AND HETEROLOGOUS 7S GAMMA GLOBULIN FRAGMENTS. J Exp Med. 1965 Mar 1;121(3):323-38. doi: 10.1084/jem.121.3.323. PMID: 14271318.

8) Potocnjak P, Yoshida N, Nussenzweig RS, Nussenzweig V. Monovalent fragments (Fab) of monoclonal antibodies to a sporozoite surface antigen (Pb44) protect mice against malarial infection. J Exp Med. 1980 Jun 1;151(6):1504-13. doi: 10.1084/jem.151.6.1504. PMID: 6991628.

9) Mackness BC, Jaworski JA, Boudanova E, Park A, Valente D, Mauriac C, Pasquier O, Schmidt T, Kabiri M, Kandira A, Radošević K, Qiu H. Antibody Fc engineering for enhanced neonatal Fc receptor binding and prolonged circulation half-life. MAbs. 2019 Oct;11(7):1276-1288. doi: 10.1080/19420862.2019.1633883. PMID: 31216930

10) Ko S, Park S, Sohn MH, Jo M, Ko BJ, Na JH, Yoo H, Jeong AL, Ha K, Woo JR, Lim C, Shin JH, Lee D, Choi SY, Jung ST. An Fc variant with two mutations confers prolonged serum half-life and enhanced effector functions on IgG antibodies. Exp Mol Med. 2022 Nov;54(11):1850-1861. doi: 10.1038/s12276-022-00870-5. Epub 2022 Nov 1. PMID: 36319752.

---

## [Editor Report · Decision Letter 1]

21 Mar 2023

PONE-D-22-23114R1Transport of Antibody into the Skin is Only Partially Dependent Upon the Neonatal Fc-ReceptorPLOS ONE

Dear Dr. Sinnis,

Thank you for submitting your manuscript to PLOS ONE. After careful consideration, we feel that it has merit but does not fully meet PLOS ONE’s publication criteria as it currently stands. Therefore, we invite you to submit a revised version of the manuscript that addresses the points raised during the review process. I am sorry for not being clear enough in my previous comment. The following sentence in the abstract is misleading: "To understand the relevance of FcRn in the context of malaria infection, we use the rodent parasite *Plasmodium berghei* and show that the protective efficacy of a passively-administered anti-malarial antibody is reduced in FcRn deficient mice, but not to the same extent as previously observed in wildtype mice." The way I understand this sentence is that protective efficacy of the antibody is less reduced in FcRn deficient mice as compared to WT mice, which is the opposite of what the data show. The parasite load is less reduced, not the protective efficacy. The protective efficacy is reduced as compared to WT.

We look forward to receiving your revised manuscript.

Kind regards,

Olivier Silvie, M.D., Ph.D.

Academic Editor

PLOS ONE
---

## [Author Response · Author response to Decision Letter 1]

28 Mar 2023

We agree that this sentence was confusing and have now changed the sentence to read:

To understand the relevance of FcRn in the context of malaria infection, we use the rodent parasite Plasmodium berghei and show that passively-administered anti-malarial antibody in FcRn deficient mice, does not reduce parasite burden to the same extent as previously observed in wildtype mice.

---

## [Editor Report · Decision Letter 2]

30 Mar 2023

Transport of Antibody into the Skin is Only Partially Dependent Upon the Neonatal Fc-Receptor

PONE-D-22-23114R2

Dear Photini,

We’re pleased to inform you that your manuscript has been judged scientifically suitable for publication and will be formally accepted for publication once it meets all outstanding technical requirements.

Kind regards,

Olivier Silvie, M.D., Ph.D.

Academic Editor

PLOS ONE
---

## [Editor Report · Acceptance letter]

13 Apr 2023

PONE-D-22-23114R2 

Transport of Antibody into the Skin is Only Partially Dependent Upon the Neonatal Fc-Receptor 

Dear Dr. Sinnis:

I'm pleased to inform you that your manuscript has been deemed suitable for publication in PLOS ONE. Congratulations! Your manuscript is now with our production department. 

Kind regards, 

on behalf of

Dr Olivier Silvie 

Academic Editor

PLOS ONE